# Co_3_Gd_4_ Cage as Magnetic Refrigerant and Co_3_Dy_3_ Cage Showing Slow Relaxation of Magnetisation

**DOI:** 10.3390/molecules27031130

**Published:** 2022-02-08

**Authors:** Javeed Ahmad Sheikh, Himanshu Sekhar Jena, Sanjit Konar

**Affiliations:** 1Department of Chemistry, Government, College for Women, Constituent College of Cluster University, M. A. Road, Srinagar 190001, Jammu and Kashmir, India; 2Department of Chemistry, IISER Bhopal, Bhopal By-Pass Road, Bhopal 462066, Madhya Pradesh, India; himanshu.jena@ugent.be or; 3Department of Chemistry, Ghent University, Krijgslaan 281-S3 B, 9000 Ghent, Belgium

**Keywords:** coordination cluster, single molecular magnet, magnetic refrigerant, cages

## Abstract

Two structurally dissimilar 3d-4f cages having the formulae [(Co^III^)_3_Gd_4_(μ_3_-OH)_2_(CO_3_) (O_2_C^t^Bu)_11_(teaH)_3_]·5H_2_O (**1**) and [(Co^III^)_3_Dy_3_(μ_3_-OH)_4_(O_2_C^t^Bu)_6_(teaH)_3_]·(NO_3_)_2_·H_2_O (**2**) have been isolated under similar reaction conditions and stoichiometry of the reactants. The most important factor for structural diversity seems to be the incorporation of one μ_3_-carbonate anion in **1** and not in **2**. Co atoms are in a +3 oxidation state in both complexes, as shown by the Bond Valence Sum (BVS) calculations and bond lengths, and as further supported by magnetic measurements. Co_3_Gd_4_ displays a significant magnetocaloric effect (−∆S_m_ = 25.67 J kg^−1^ K^−1^), and Co_3_Dy_3_ shows a single molecule magnet (SMM) behavior.

## 1. Introduction

The study of nanoscopic paramagnetic metal-ion aggregates has excelled in the last decade or so, not only because of aesthetically pleasing structures but also because of their potential technological applications such as quantum computing [1,2,3], ultra-high-density data storage [4,5,6,7,8,9], molecular spintronics [10] and magnetic refrigeration [11,12,13,14,15]. In the field of magnetochemistry, molecular nanomagnets based on Gadolinium show magnetic refrigeration based on the magnetocaloric effect (MCE) [16,17,18,19]. Some of the molecular magnetic aggregates that are particularly based on Dysprosium are also useful as single-molecule magnets (SMMs) [20,21,22,23,24,25,26,27,28,29,30].

Polymetallic 3d-4f systems have developed as a fascinating sub-area of research in magnetism [31,32,33,34,35,36,37,38,39,40,41,42,43,44,45]. Accordingly, a large number of multinuclear 3d-4f cages have been reported in the literature, mainly with carboxylate, tripodal alkoxides and related ligands [31,35,36,37,38,39,40]. The inclusion of 4f ions (e.g., Dy^3+^, Tb^3+^, etc.) with 3d metals in nanosized systems (3d–4f approach) has been used to incorporate a large number of unpaired f electrons and high intrinsic magnetic anisotropy to obtain SMMs. SMMs based on Dy^III^ exceed those of other Ln^III^-based SMMs, most likely due to their larger *m*_J_ state (*m*_J_ = ±15/2), which could lead to an appreciable magnetic moment. Further, Dy^III^ is a Kramer’s ion (it has an odd number of f-electrons), indicating that the ground state will always be bistable irrespective of the crystal field symmetry. Gd^3+^ ions are suitable for the MCE, as they have a high isotropic spin, quenched orbital momentum and weak superexchange interactions. Therefore, 3d-Ln cages comprised of Gd^III^ ions are suitable as magnetic refrigerants [11,12,13,14,15,46,47,48], and those with Dy^III^ ions (anisotropic) are ideal for SMM behaviour [31,32,33,34,35,36,37,38,39,40,41,42,43,44,45].

Alkoxo ligands such as N-substituted diethanolamines have been widely used in the synthesis of 3d-4f cages with different numbers of metal centers and magnetic properties [49,50,51,52]. Recently, we have reported a series of complexes based on the N-n-butyldiethanolamine ligand [53]. With the intention of obtaining new heterometallic cages with an increased magnetic density by decreasing the content of non-metallic elements, we used an alkoxo ligand containing more hydroxyl groups (triethanolamine).

## 2. Results

### 2.1. Synthesis and Structural Analysis

In this work, we have employed triethanolamine (teaH_3_) with the aim of obtaining new heterometallic cages. The use of the said ligand for such systems is rare [54,55,56,57]. We have reacted it with a small dimer, [Co_2_(μ-OH_2_) (O_2_C^t^Bu)_4_]·(HO_2_C^t^Bu)_4_ (hereafter: Co_2_), and lanthanide salts, and have successfully isolated two complexes with different structural features under similar reaction conditions and stoichiometry of the reactants. The reaction of Co_2_ and teaH_3_ with Gd(NO_3_)_3_·6H_2_O in a 1:1:1 molar ratio in CH_3_CN gave the compound [(Co^III^)_3_Gd_4_(μ_3_-OH)_2_(CO_3_) (O_2_C^t^Bu)_11_(teaH)_3_]·5H_2_O (**1**), whereas a similar reaction with Dy(NO_3_)_3_·6H_2_O led to the compound [(Co^III^)_3_Dy_3_(μ_3_-OH)_4_(O_2_C^t^Bu)_6_(teaH)_3_]·(NO_3_)_2_·H_2_O (**2**).

X-ray crystallography reveals that complex **1** crystallises in the *P*-1 space group and is a heterometallic heptanuclear cage primarily composed of three cobalt centres and four Gd^III^ ions (Figure 1). All three cobalt centres are in a +3 oxidation state, as shown by the Bond valence sum (BVS) calculations and bond lengths [58]. Two μ_3_-hydroxo groups and oxygen atoms of one carbonate anion interconnect the metal centres of this heptanuclear core (Figure 2). Peripheral ligation is provided by three doubly deprotonated triethanolamine ligands (teaH). One can observe that N atoms coordinate to the cobalt ions and that oxygens coordinate to the Gd^III^ ions. The central portion is also enveloped by a hydrophobic covering of eleven pivalate ligands bridging in the 2.11 mode. Therefore, all the cobalt ions end up with an octahedral geometry (with O5N coordination), and the Gd^III^ ions feature a distorted square antiprismatic geometry. The average Co^III^−O and Co^III^−N bond lengths are 1.90 (5) and 1.98 (6) Å, respectively, and the average Gd−O bond length is 2.37 (5) Å.

Complex **2** crystallises in the monoclinic space group *P*21/c and is another example of a mixed metal system comprising three Co^III^ and three Dy^III^ ions (Figure 3).

The metal centres and oxygen atoms in the central hexanuclear unit are interlinked in a hemicubane-like fashion by four μ_3_-hydroxo groups (Figure 4). Three doubly deprotonated triethanolamine ligands (teaH) are also part of this structural aggregation, coordinating via the N atom to the Co^III^ ions and then bridging the Co^III^ centres to the Dy^III^ ions via their two μ_2_-alkoxo groups. Six pivalate groups bridging in the 2.11 mode and three water molecules, one each coordinating to the Dy^III^ ions, also surround the basic unit. With all these coordinating atoms of the ligands, the Co^III^ ions end up being six-coordinated with an octahedral geometry having average Co−O and Co−N bond distances of 1.90 (1) and 1.97 (2) Å, respectively. All the Dy^III^ ions are eight-coordinated, having a distorted square antiprismatic geometry and average Dy−O bond length of 2.36 (6) Å. The average Co−O−Dy and Dy−O−Dy bond angles are 102 (3)⁰ and 110 (3)⁰, respectively.

The careful study of the two structures (**1** and **2**) shows some interesting features. Although the number of chelating teaH ligands is the same in both complexes, the number of pivalate groups is reduced to almost half from the former to the latter complex (from 11 in **1** to 6 in **2**). The incorporation of one μ_3_-carbonate anion seems the most important factor for this structural diversity. The number of μ_3_-hydroxo groups is also doubled from two to four from the former to the latter complex, respectively. One μ_3_-hydroxo group is found bridging the three Dy^III^ centres in complex **2** and is not present in complex **1**. Another difference is that each of the Dy^III^ centres in the latter complex is coordinated to one water molecule to complete the coordination sphere.

### 2.2. Magnetic Studies

Polycrystalline samples of **1** & **2** were used to collect the dc susceptibility data in the temperature range of 1.8–300 K at 0.1 T. The DC magnetic studies (Figure 5) reveal room temperature *χ*_M_T values of 30.56 and 42.41 cm^3^ mol^−1^ K for **1** and **2**, respectively, which are quite close to the estimated values of 29.24 (four uncoupled Gd^III^, g = 1.99) and 42.71 (**2b**, three uncoupled Dy^III^, g = 4/3). Upon lowering the temperature, the *χ*_M_*T* products stay nearly constant for complex **1** up to 40 K, where an abrupt decrease is witnessed, reaching a value of 22.89 cm^3^ mol^−1^ K at 0.1 T and 1.8 K. This behaviour can be ascribed to the isotropic nature of the Gd^III^ ions. For complex **2**, upon lowering the temperature, the *χ*_M_*T* values are nearly constant up to 60 K, followed by an abrupt decrease, reaching a value of 15.75 cm^3^ mol^−1^ K at 0.1 T and 1.8 K. This fall could be due to the depopulation of the Stark (m*_J_*) sublevels of the ground J multiplet, with the likelihood of a feeble antiferromagnetic exchange and dipolar interactions also backing the behaviour.

For complex **1**, the field dependence of magnetisation shows a saturation value of 28.8 Nμ_B_ at 7 T (Appendix A). This is compatible with the predicted value of 28 Nμ_B_. The entropy variations (∆S_m_) for **1** were estimated using the Maxwell equation ∆S_m_(T)_∆H_ = ∫[∂M(T,H)/∂T]_H_dH [59]. The −∆S_m_ vs. T plot gradually increases from 9 K to 2 K (Figure 6), reaching a maximum of 25.67 J kg^−1^ K^−1^ at 3 K and 7 T. These results compare well with the other Co−Gd cages in the literature [60,61,62,63].

The M/Nμ_B_ vs. H plot for **2** (Appendix A) shows an abrupt increase with the increasing field reaching a value of 17.28 Nμ_B_ but not saturating, even at 7 T. This is usually due to the presence of anisotropy and significant crystal field effects from the Dy^III^ ions [54,55,56,57]. The non-superposition of the M/Nμ_B_ versus H/T plot of complex **2** (Appendix A) confirms the presence of significant anisotropy in the molecule.

Alternating current susceptibility data for **2** was collected at a zero dc field up to 1.8 K at the 3.5 Oe ac field in the frequency range of 1–800 Hz. Both the in-phase and out-of-phase susceptibilities show temperature-dependent ac signals below 10 K (Appendix A), indicating the slow relaxation of magnetisation. Due to quantum tunnelling of the magnetisation (QTM), no full maxima were observed [64,65]. The data was remeasured in the presence of an optimum static dc field of 2000 Oe to minimise the quantum tunnelling. Peak maxima were observed under this field below 5 K in the out-of-phase (*χ″*) vs. T plot (Figure 7 (left)), confirming the field-induced SMM behaviour [54,55,56,57]. The frequency-dependent in-phase (*χ′*) and out-of-phase (*χ″*) susceptibility plots also confirmed this behaviour (Appendix A and Figure 7 (right)). The magnetic properties of this compound strongly resemble one of our previously reported compounds because of the similar core structure [53].

The best-fitting results for the Arrhenius equation (Equation (1)) [66,67] gave an energy barrier U_eff_ ≈ 17.5 K and a relaxation time τ_0_ ≈ 2.3 × 10^−6^ s from the frequency dependencies of the ac susceptibility (Figure 8).
ln(1/τ) = ln(1/τ_0_) − U_eff_/kt(1)
where k is the Boltzmann constant, and 1/τ_0_ is the pre-exponential factor.

The Cole−Cole plot (*χ″* vs. *χ′*) is shown in the inset of Figure 8 as evidence of the relaxation process occurring in complex **2**.

## 3. Materials and Methods

Both complexes were synthesised from the starting material [Co_2_(μ-OH_2_)(O_2_C^t^Bu)_4_]·(HO_2_C^t^Bu)_4_, Co_2_. All the reagents were used as received from Sigma Aldrich without any further purification. The magnetic behaviour of the compounds was studied on a Quantum Design SQUID-VSM magnetometer. Diamagnetic corrections were made with Pascal’s constants for all of the constituent atoms [68]. Magnetic susceptibility measurements were performed in 1.8–300 K with an applied field of 0.1 T. Infrared spectra were collected for the solid samples using KBr pellets on a Perkin Elmer Fourier-transform infrared (FTIR) spectrometer in the range of 400–4000 cm^−1^. An Elementar vario Microcube elemental analyser was used to get the elemental analysis data.

Single-crystal X-ray structural studies of **1** & **2** were carried out on a CCD Bruker SMART APEX 2 CCD diffractometer under the cold flow of an Oxford device. Data were collected using graphite−monochromated Mo Kα radiation (λα = 0.71073 Å). Structure solution, refinement and data reduction were carried out by (SHELXTL-97), SAINT and SADABS programs [69,70,71]. Large solvent accessible voids are present in the structures, which are probably filled with disordered solvent molecules. Therefore, SQUEEZE/PLATON was used to remove or fix these disorders [72]. The CIF format of the data is available in CCDC numbers 1,050,639 and 1,050,640 and is also summarized in Appendix A.

## 4. Conclusions

Two structurally dissimilar heterometallic aggregates were successfully synthesised from a preformed precursor and triethanolamine. The Gd analogue displays a significant magnetocaloric effect, and the Dy-containing compound shows the slow relaxation of the magnetisation. The results are a good addition to the 3d-4f heterometallic aggregates in general and those obtained from polyalcohol-based ligands in particular. This work should be useful to the sensible strategy and production of a library of heterometallic magnetic materials employing different polytopic ligands.

## Figures and Tables

**Figure 1 molecules-27-01130-f001:**
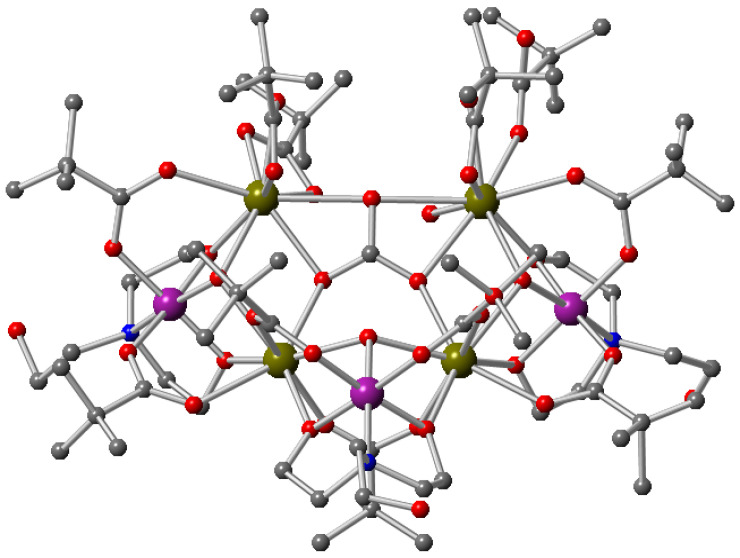
Molecular structure of **1** in the crystal. Colour code: purple, Co^III^; olive, Gd^III^; red, oxygen; blue, nitrogen; grey, carbon; Hydrogen atoms are omitted for clarity.

**Figure 2 molecules-27-01130-f002:**
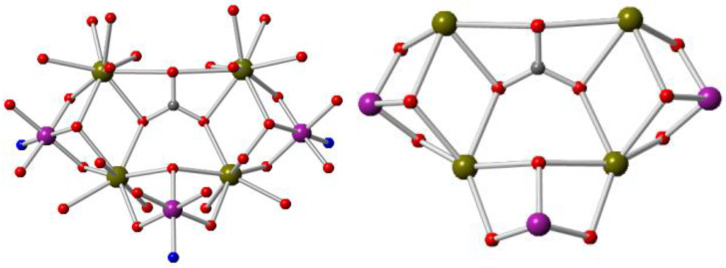
Core structure of **1** (**left**). Colour code: same as Figure 1. View of the fine core (**right**).

**Figure 3 molecules-27-01130-f003:**
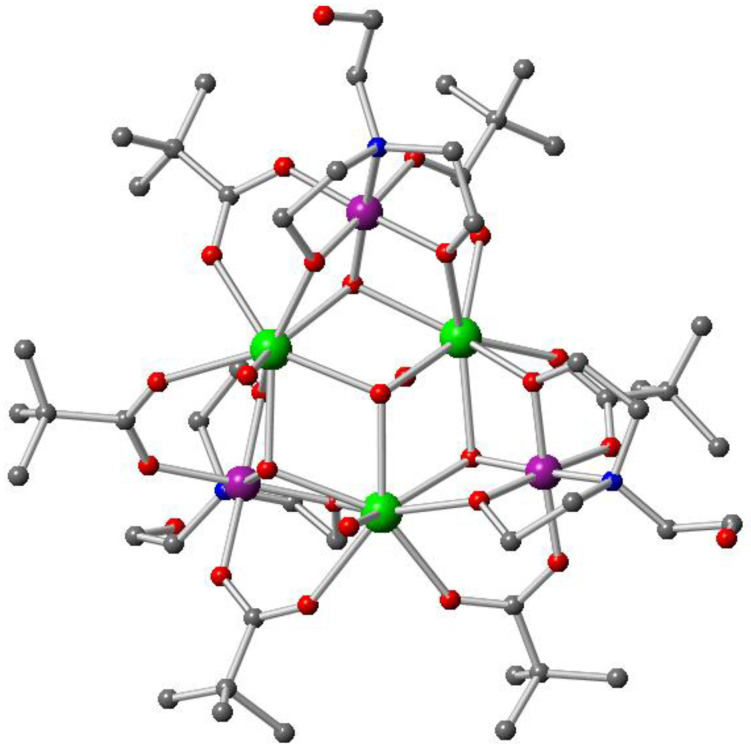
Molecular structure of **2** in the crystal. Colour code: purple, Co^III^; green, Dy^III^; red, oxygen; blue, nitrogen; grey, carbon; Hydrogen atoms are omitted for clarity.

**Figure 4 molecules-27-01130-f004:**
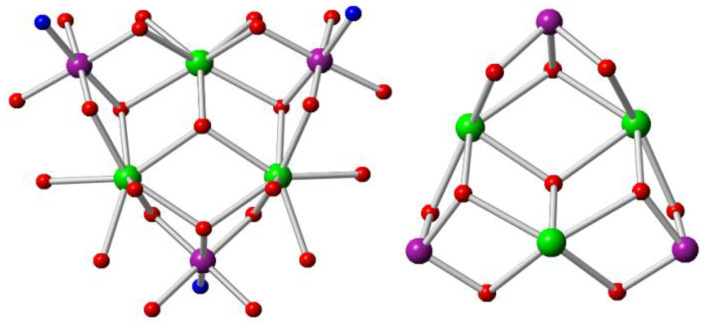
Core structure of **2** (**left**) and hemicubane-like view of the core (**right**) Colour code: purple, Co^III^; green, Dy^III^; red, oxygen; blue, nitrogen; grey, carbon; Hydrogen atoms are omitted for clarity.

**Figure 5 molecules-27-01130-f005:**
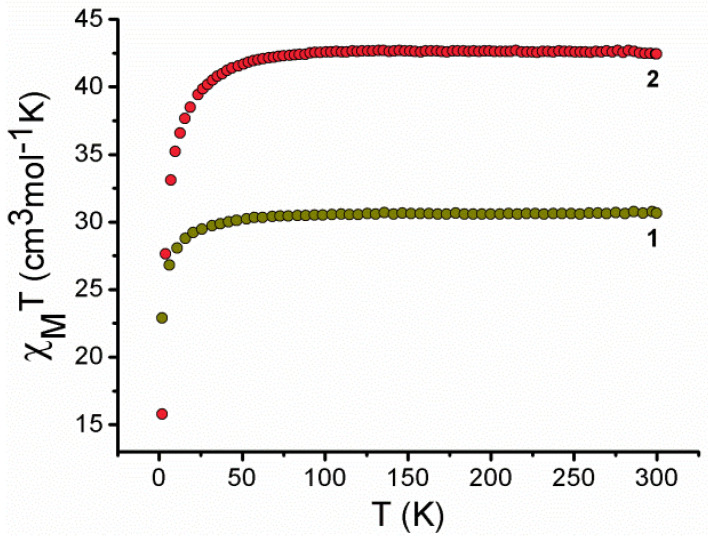
Temperature dependence of *χ*_M_T measured at 0.1 T for complexes **1** and **2**.

**Figure 6 molecules-27-01130-f006:**
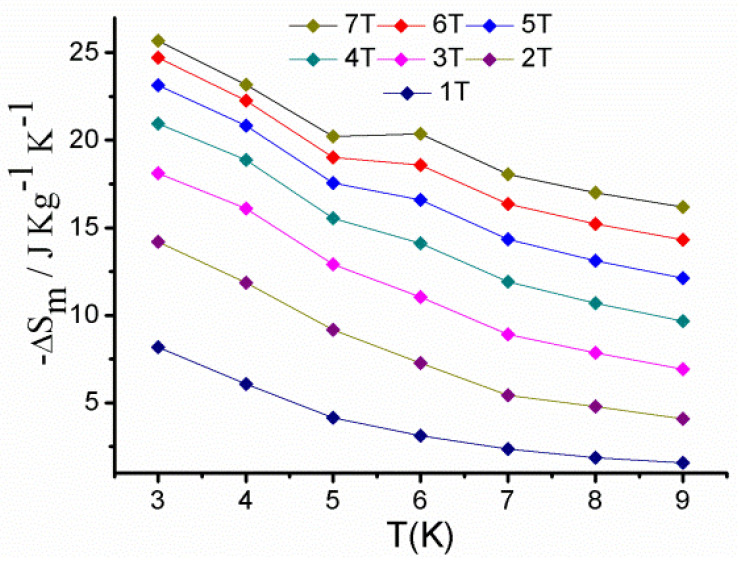
Temperature dependencies (3 to 10 K) of the magnetic entropy change (−ΔS_m_) for complex **1**, as obtained from magnetisation data.

**Figure 7 molecules-27-01130-f007:**
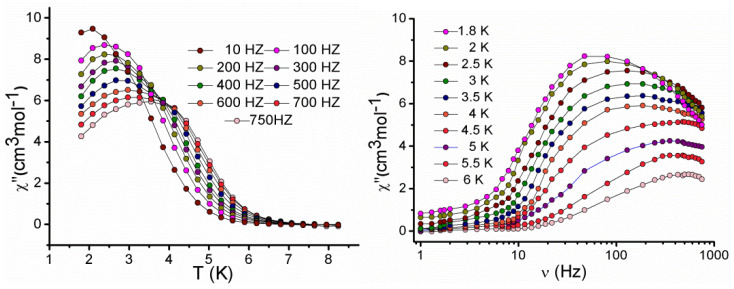
Temperature dependence (**left**) and frequency dependence (**right**) of the out−of−phase *χ″* ac susceptibility for complex **2** under a 2000 Oe dc field.

**Figure 8 molecules-27-01130-f008:**
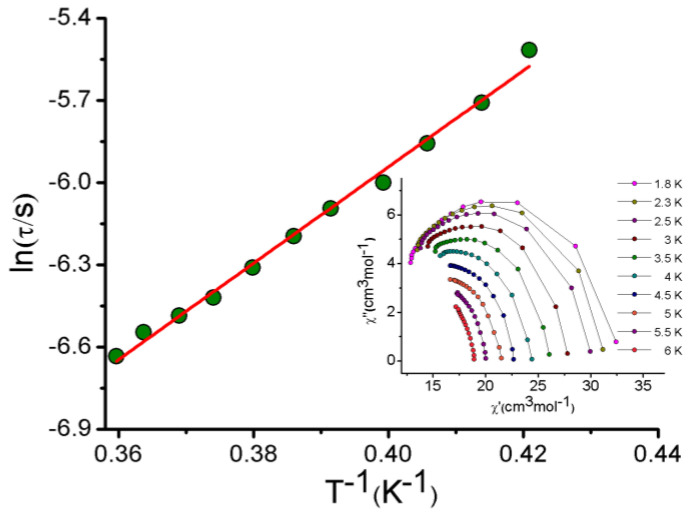
ln (1/τ) vs. 1/*T* plot for complex **2**. The red line is the best fit for the Arrhenius relationship.

## Data Availability

This study did not report any data.

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
