# Peer review of "Co_3_Gd_4_ Cage as Magnetic Refrigerant and Co_3_Dy_3_ Cage Showing Slow Relaxation of Magnetisation"

_molecules, 2022, doi:10.3390/molecules27031130_

Round 1

Reviewer 1 Report

The authors synthesized two structurally dissimilar heterometallic aggregates by situ synthetic route. In my opinion, the paper is suitable to be published in molecules after revised the manuscript. Some notable comments are as following.

  1. References should be cited in the past five years.
  2. The novelty should be better discussed in the present manuscript.
  3. In the abstract more information about results should be added.

Author Response

The authors synthesized two structurally dissimilar heterometallic aggregates by situ synthetic route. In my opinion, the paper is suitable to be published in molecules after revised the manuscript. Some notable comments are as following.

  • References should be cited in the past five years.

Answer: We have included additional references from the past 5 years at relevant places. Please see the track changes.

  • The novelty should be better discussed in the present manuscript.

Answer: we have added the following sentence in the introduction,

With the intention to obtain new heterometallic cages with increased magnetic density by decreasing the content of non-metallic elements, we used alkoxo ligand containing more hydroxyl groups (triethanolamine).

  • In the abstract more information about results should be added.

Answer: Thank you for your valuable suggestion. We have modified the abstract and included more information as follows,

Two structurally dissimilar 3d-4f cages having formulae, [(CoIII)3Gd43-OH)2(CO3) (O2CtBu)11(teaH)3]·5H2O (1) and [(CoIII)3Dy33-OH)4(O2CtBu)6(teaH)3]·(NO3)2∙H2O (2) have been isolated under similar reaction conditions and stoichiometry of the reactants. The most important factor for the structural diversity seems to be the incorporation of one μ3-carbonate anion in 1 and not in 2.  Co atoms are in +3 oxidation state in both the complexes as shown by Bond Valence Sum (BVS) calculations, bond lengths and further supported by magnetic measurements. Co3Gd4 displays significant magnetocaloric effect (-∆Sm = 25.67 J kg-1 K-1) and Co3Dy3 shows single molecule magnet (SMM) behavior.

Reviewer 2 Report

The manuscript titled Co3Gd4 cage as magnetic refrigerant and Co3Dy3 cage showing slow relaxation of magnetization presents the synthesis and characterization of two new molecules with detectable magnetic properties at low temperatures and possible application in extreme cooling. The manuscript findings are incremental, suitable for a communication. The complexes are investigated by standard techniques. The work is original, but the abstract and the introduction doesn’t enclose sufficient information, and the conclusion is too general. References are adequate and all illustrations and tables are necessary or present in the supplementary file.

Author Response

The manuscript titled Co3Gd4 cage as magnetic refrigerant and Co3Dy3 cage showing slow relaxation of magnetization presents the synthesis and characterization of two new molecules with detectable magnetic properties at low temperatures and possible application in extreme cooling. The manuscript findings are incremental, suitable for a communication. The complexes are investigated by standard techniques. The work is original, but the abstract and the introduction doesn’t enclose sufficient information, and the conclusion is too general. References are adequate and all illustrations and tables are necessary or present in the supplementary file.

Answer:

We are highly obliged by the encouraging comments of the referee. We have modified the abstract as follows,

Two structurally dissimilar 3d-4f cages having formulae, [(CoIII)3Gd43-OH)2(CO3) (O2CtBu)11(teaH)3]·5H2O (1) and [(CoIII)3Dy33-OH)4(O2CtBu)6(teaH)3]·(NO3)2∙H2O (2) have been isolated under similar reaction conditions and stoichiometry of the reactants. The most important factor for the structural diversity seems to be the incorporation of one μ3-carbonate anion in 1 and not in 2.  Co atoms are in +3 oxidation state in both the complexes as shown by Bond Valence Sum (BVS) calculations, bond lengths and further supported by magnetic measurements. Co3Gd4 displays significant magnetocaloric effect (-∆Sm = 25.67 J kg-1 K-1) andCo3Dy3 shows single molecule magnet (SMM) behavior.

The following sentences have been added to the introduction,

SMMs based on DyIII exceed that of other LnIII based SMMs, most likely due to its larger mJ state (mJ = ±15/2), that could lead to an appreciable magnetic moment. Further, DyIII is a Kramer’s ion (it has an odd number of f-electrons), indicating that the ground state will always be bistable irrespective of the crystal field symmetry. Gd3+ ions are suitable for MCE as it has high isotropic spin, quenched orbital momentum, and weak superexchange interactions.

With the intention to obtain new heterometallic cages with increased magnetic density by decreasing the content of non-metallic elements, we used alkoxo ligand containing more hydroxyl groups (triethanolamine).

The conclusion has also been modified as follows,

Two structurally dissimilar heterometallic aggregates were successfully synthesised from a preformed precursor and triethanolamine. The Gd analogue displays significant magnetocaloric effect and the Dy containing compound shows slow relaxation of the magnetization. The results are a good addition to the 3d-4f heterometallic aggregates in general and those obtained from polyalcohol based ligands in particular. This work should be useful in the sensible strategy and production of a library of heterometallic magnetic materials by employing different polytopic ligands

Round 2

Reviewer 2 Report

The authors addressed in the revised manuscript all issues raised by the reviewer.